# Research into the Effect of Grain and the Content of Alundum on Tribological Properties and Selected Mechanical Properties of Polymer Composites

**DOI:** 10.3390/ma13245735

**Published:** 2020-12-16

**Authors:** Aneta Krzyzak, Ewelina Kosicka, Robert Szczepaniak

**Affiliations:** 1Faculty of Aeronautics, Military University of Aviation, 08-521 Deblin, Poland; r.szczepaniak@law.mil.pl; 2Faculty of Mechanical Engineering, Lublin University of Technology, 20-618 Lublin, Poland; e.kosicka@pollub.pl

**Keywords:** polymer composites, tribology, abrasive wear, alundum, epoxy resin

## Abstract

The subject of the research is a polymer composite with a matrix base of epoxy resin L285 cured with H285 hardener, and a physical modifier of friction in the form of alundum. The article presents an analysis of findings of tribological examinations. The authors evaluated the influence of the modifier properties in the form of alundum, i.e., mass share and grain size, on the abrasive wear of a composite, defined as loss of weight as well as on roughness parameters and selected mechanical properties. The tribological examinations have been extended by measurements of hardness and density of the prepared composites. The obtained results of tribological examinations showed an increase in the average value of weight loss in relation to the loss of sample weight loss between the cycles. The influence of both the grain size and the mass percentage share of alundum upon the increase in the longitudinal modulus of elasticity was also observed. On the basis of the obtained results, it was found that alundum of grain sizes equal to F220 and F240 exerted the best influence on the reduction of abrasive wear of the tested samples. In the case of F220, it was 14.04% of the average value of the weight loss between the cycles for all percentage shares of the used grains.

## 1. Introduction

Modern industries, especially mechanical engineering [1] and the automotive one [2], are constantly searching for construction materials, which may, due to their enhanced useful properties and increased durability, replace the materials used so far [3,4]. Continuous progress in the field of material engineering provides an opportunity to produce new or improved structural materials, which must undergo extensive research before their production can be optimized and approved for use [5]. The selection of material for a specific constructional application requires knowledge of its physical and chemical properties, ensuring that it will be susceptible to the required shaping to obtain the finished product. Furthermore, the material should meet both economic and environmental criteria (for future recycling) [6]. The simulations [7,8] and laboratory tests [9,10,11] comprehensively conducted over the manufactured engineering materials result from their later use in components of responsible structures of machinery or devices [12].

Composite materials have long been used due to [13,14]:
–good constructional properties,–small specific gravity,–easy moulding of products (also those of large dimensions),–diversity of processing techniques,–possibility of differentiating the properties by modifying the use of intermediate inputs and processing techniques.

In addition, their use was popularized by the fact that it was possible to introduce changes to the structural properties by including modifiers in their composition [15,16].

When classifying modifiers by origin, it is possible to distinguish [17,18,19,20]:–natural organic fillers (e.g., wood flour, flax fibres),–inorganic fillers(e.g., chalk, talc, quartz),–synthetic fillers (e.g., glass fibres, carbon fibres).

Taking into consideration the form in which the fillers appear, the following can be distinguished:–powder fillers (spherical, flake, or short cut fibre form),–fibre fillers (e.g., glass fibres, boron fibres).

Physical modifiers and reinforcing materials improve material properties such as density, hardness, impact strength, and structural properties, including the tribological ones [21,22].

The problems of tribological wear and tear of machine and equipment components ultimately translate into their performance and operational reliability. The occurrence of numerous machine operational units makes it necessary to limit this irreversible phenomenon, which is achieved, for example, by using construction materials with improved anti-wear properties. Conducting tribological research in industrial conditions is difficult and costly [23]. Therefore, an analysis of friction and wear processes is carried out on designed friction devices (interchangeably called tribometers or tribotesters). Their task is to reproduce friction through model experiments. Thus, they differ both in design and manner of operation. The most frequently measured values, during such tests, are as follows:–wear rate,–linear or volumetric wear,–force of friction,–coefficient of friction,–temperature of the contact area.

A literature analysis related to tribological issues repeatedly refers to automotive-related research [24,25]. The development of science with regard to material abrasion is also developing in the field of biomedical engineering, including examples of hip and knee joint implants [26,27] or dental materials [28,29].

Aeronautics is one of the areas of application of construction materials in which tribological aspects are subject to an in-depth analysis. This is influenced by specific operating conditions of manufactured components combined with high safety requirements. The available literature presents the research results on selected areas that require improvement in friction parameters. They relate, among others, to aircraft brakes, e.g., of the carbon-carbon type (research into the effect of infiltration of silicon carbide nano-additives on a porous composite used to manufacture brake discs [30]), containing carbon/silicon carbide [31], or reinforced with carbon fibres using various manufacturing technologies [32]. By emphasising the need to improve tribological properties in this area, through the conducted research, it was possible to obtain an increase in the effective coefficient of friction by almost 60% with the presence of approximately 0.6% mass share of Si in the composite.

In addition to the areas indicated, the authors of the research publish the obtained findings with regard to improved properties of aircraft skin. In this case, it is important to take into account a wide range of forces occurring in operation, also including extreme operating conditions which result from humidity changes depending on a changing altitude, a wide range of temperature values, and the occurrence of abrasive wear caused by electrostatic attraction of dust. Improvement in tribological properties of composite materials can be conducted by plasma spraying [33] or the use of physical friction modifiers [34,35].

The research part of this publication deals with an inorganic powder filler in the form of an abrasive, i.e., alundum of different grain sizes. The granularity of abrasives is determined on the basis of the sieve analysis (up to 53 μm). The digital marking used in the notation indicates the number of holes found in a specified unit of the sieve area. The thickest grains are identified by the symbol F8, whereas the finest ones by F1200. Apart from the characteristics of the abrasives in the form of described grain, their hardness is often taken into consideration. For example, granite has the hardness of approximately 14 GPa, quartz of approximately 11 GPa, while the hardness of alundum equals 20–22 GPa on the Vickers scale [36].

In the available literature, it is possible to find numerous examples of research conducted so far in which an influence of various fillers that acted as friction modifiers was determined. The authors checked the influence of various mass percentage shares of graphite on tribological properties [37,38,39,40] in a composite material. Other carbon structures such as the obtained carbon nanotubes or glassy carbon [41] are also of interest [38,42,43] (Table 1).

In the available literature, among other friction modifiers, it is possible to find, e.g.: polyphenylene sulphide (PPS) [44], TiO_2_ [45,46], or Al_2_O_3_ [47]. Research is being conducted, more and more frequently, into the tribological properties of natural fibres [48] such as bamboo fibre [49], sugarcane fibers [50], or jute fibre [51]. It is worth adding that apart from natural fibres, natural nanopowders are also used [52].

The cited publications, which present only a narrow slice of the variety of areas of tribological research, prove that the search for dependencies describing the processes of material consumption is still up to date. The observed intensification of the development of materials engineering, apart from defining the obtained mechanical properties for a given material, also enforces determining tribological properties, taking into account further possibilities of exploiting these materials in construction solutions, which include friction pairs. It is, therefore, appropriate to identify the existing dependencies that may, in the long term, determine the application of new solutions in practice or indicate operational recommendations.

## 2. Methodology of Research

The research was carried out using polymer composites with a matrix base, whose reinforcement phase was alundum (EA symbol) Al_2_O_3_, 99% pure, and with F220, F240, F280, F320, F360 grains (the division of abrasive grain in accordance with FEPA 42-2:2006). The actual grain size, in accordance with the standard FEPA is, respectively: 53, 44.5, 36.5, 29.2, and 22.8 µm [53].

The designation of the fraction as F0 indicates pure epoxy resin, without the addition of alundum. The examinations were made in the laboratory of the Department of Airframe and Engine at the Military University of Aviation in Dęblin (Poland). For the preparation of samples by gravity casting, epoxy resin with the trade name L285, along with H285 MGS hardener, were used, as well as alundum with mass percentage share of 5%, 10%, 15%, 20%, and 25%, respectively, for each of the mentioned grains. Gravity casting consisted of preparing a mixture of resin, hardener, and EA, and then filling the moulds with it. After removing the air bubbles using ultrasonic waves, the mould was allowed to stand for 24 h at room temperature until the resin was completely cured. Each batch contained 10 samples and the results were averaged. In order to determine whether the grain distribution was homogeneous, the fracture surface was observed on the optical microscope OLIMPUS BX53M+ (Olympus Corporation, Tokyo, Japan). Figure 1 shows two exemplary fracture surfaces. There is no agglomerate of alundum.

The produced samples were tested in conditions of abrasive wear in a reciprocating motion using the tribotester—Taber Linear Abraser, model 5750 (North Tonawanda, NY, USA). The abrasive stone, used on the test stand, was 6.6 mm in diameter. Its grain equalled 200. The friction path was 177.8 mm/cycle and the total load equalled 1850 g. Prior to the friction process, the samples were weighed. Additionally, their initial weight was determined. After 100, 300, 600, and 1000 cycles, the weight of the samples was determined again using the precision laboratory balance XSE205DU/M (Mettler Toledo, Zurich, Switzerland). After each measurement, the sample was carefully cleaned, also removing the products of friction. Moreover, in accordance with the manufacturer’s recommendations, the abrasive stone and the sample were cleaned. After each interval of cycles, the authors measured the selected roughness parameters in a cross-section, perpendicular to the direction of the path covered by an anti-sample. The measurement of the roughness parameters, Ra (arithmetical mean roughness value) and Rmax (roughness depth), was made by means of an optical profilometer (MicroProf 100 FRT, FRT GmbH, Bergisch Gladbach, Germany). Moreover, in accordance with the PN-EN ISO 527-2:2012 [54] norm, a static tensile test was conducted using a Zwick Z5.0 TN ZwickLine testing machine (ZwickRoell AG, Ulm, Germany). The elongation was carried out with the following parameters: measuring length of 90 mm, the traverse movement equal to 2 mm/min.

In the further part of the research programme, the samples were subjected to hardness measurements in accordance with the Shore D method at a three-second load of 5000 g, in compliance with the PN-ISO 868 [55] norm using the hardness tester Digi Test II, Type DTAA Bareiss (FRT GmbH, Bergisch Gladbach, Germany). In addition, for each composite of different grain alundum and its percentage value, the authors determined density by means of the gravimetric method using an analytical balance XSE205DU/M (Mettler Toledo, Zurich, Switzerland). 

## 3. Results

### 3.1. Tribological Properties

Figure 2 shows examples of generated graphs of weight loss discrepancies against loss of weight between the cycles for samples made of pure resin and hardener as well as with an addition of alundum, whose grain equals F220, F240, F280, F320, and F360; see sample discrepancy graphs for 25% alundum content (Figure 2). All other graphs of weight loss discrepancy are very close to one another, both in terms of mathematical descriptions of the regression function (Table 2) as well as of the surface distribution of the measuring points in the coordinate system. In most cases, a relatively high coefficient of determination r^2^ > 0.8 can be observed, which indicates good correspondence of the regression function to the obtained test results. Moreover, all the results, regardless of the type of the grain size and the amount of the filler, due to the similarity of the nature of the discrepancy, indicate high repeatability and consequently the predictability of weight losses during the friction process.

Moreover, the conducted observations focused on determining whether there is an influence of the grain size of alundum on the weight loss at a given percentage. Thus, for each percentage mass share of the physical friction modifier, a graph was generated showing weight loss after 0, 100, 300, 600, and 1000 cycles (Figure 3).

The generated graphs of weight loss from the initial state for particular percentage shares of physical friction modifiers proved that in the majority of cases (for 5%, 10%, and 15%), the greatest weight loss was observed in samples with the presence of alundum whose grain equaled F360. It was similar to the weight loss of pure resin samples. This results from the fact that the sharp-edged alundum grains (as shown in Figure 1), with a smaller particle size, more easily penetrate the fragile matrix of the composite. The sinking of the grain into the resin causes scratches and material tears. This is more evident in composites with a lower mass percentage share due to the fact that the friable matrix volume is greater and the distances between hard EA grains are greater. With high EA content, the grains are close to each other and loose EA grains torn out of the surface are less likely to hit the matrix material. Compared to other grains, it was found that alundum of grain size equal to F220 and F240 exerted the biggest influence on the reduction of abrasive wear of the tested samples. The abrasive wear was the lowest after each cycle interval (in the case of F220, and on average it equalled 14.04% compared to the average value of the weight loss between the cycles for all percentage shares of the exploited grains).

Along with the number of cycles, apart from the loss in mass, which can be seen in Figure 2, changes in roughness parameters Ra and Rmax are also noticeable (Figure 4). A longer and longer friction path, regardless of the grain size, decreased the values of parameter Ra, while increasing the parameter Rmax. The parameter Rmax for the content of EA in the composite, up to 15%, is higher in the extreme cases of the tested granularity, i.e., at F220 and F360. At 20% content of EA, this relationship is smoothed, and at 25%, the opposite tendency is observed. A more or less analogous change occurs in the value of the parameter Ra, with higher roughness values at extreme grain values being clearly visible only in the case of 5% of alundum.

The analysis of Figure 4 shows that the most advantageous composite in terms of application in friction joints is the composite containing 15% of alundum with grain size F280. For this composite, the estimated loss in mass is based on Figure 3 at the maximum number of friction cycles in the tested conditions, which is 0.02 g. Bearing in mind that the smaller grains at a higher percentage cause less weight loss, it can be stated that in the case of surface roughness (Ra parameter), this relationship has no significant impact. Greater surface roughness was observed with a small number of cycles. However, the difference between the Ra value with fewer cycles and the long friction process is relatively small and amounts to approximately 1 µm for each EA mass fraction. The longer the friction process lasted, regardless of the grain size, the more evenly the material losses occurred.

### 3.2. Selected Physical and Mechanical Properties

The research plan included the determination of hardness measurements of the prepared polymeric composites using the Shore method. The obtained findings are presented in the form of graphs (Figure 5). The graphs show the hardness values for different EA grains in the context of the same percentage mass shares. It was found that for EA samples with F220 grain size and 5% share, the hardness value is smaller by 0.570 °Sh compared to the reference samples. The highest increase in the hardness value compared to the samples made of pure resin was observed for EA F360 samples with 25% mass percentage share. In this case, the difference in hardness equals 3.43 °Sh (hardness increase by 4.24%).

In addition, an influence of the grain size of alundum was interpreted with regard to the density of the manufactured composites. The obtained results have been presented in Table 3. The impact of the presence of alundum on the density of the manufactured composites was observed, especially in the case of 25% mass share of alundum with grains equal to F280, F320, F360, obtaining 1.429 g/cm^3^, 1.418 g/cm^3^, and 1.41 g/cm^3^, respectively.

As mentioned in the research methodology, apart from the tribological properties, a static tensile test of the composites was planned. The obtained results have been successively presented in Figure 6, Figure 7 and Figure 8. On the OZ axis, there are successively such values as Young’s modulus (E, MPa), ultimate tensile stresses (σ, MPa), and relative elongation (ε, %). The OY axis indicates the grain size of alundum (F, μm), whereas the OX axis shows mass share of alundum in the composite (M, %).

The mathematical models of surfaces, which are a response of values of successive properties to grain changes and mass share, are quadratic functions calculated on the basis of the least squares method. The mathematical formulas are as follows:E = z = 2185.7489 − 0.1055x + 1.5136y + 0.0792x^2^ + 0.0488xy − 0.0039y^2^(1)
σ= z = 18.1254 + 1.8035x + 0.1258y − 0.0655x^2^ − 0.0004xy − 0.0001y^2^(2)
ε= z = 0.5926 + 0.1146x + 0.0059y − 0.0041x^2^ − 6.3224 × 10^−5^xy − 3.9782 × 10^−6^y^2^(3)

The conducted tensile testing proves that along with an increase of the mass percentage share of alundum, there was a rise in the value of longitudinal modulus of elasticity (Figure 6). The proved fluctuations in the value increase in the determined parameter were connected with a fluctuation occurring in the value graph of the determined density for grain F280. It forms the basis for determining the factor influencing such a trend. For most of the applied grains (F220, F280, F360), the highest value of the material’s modulus of longitudinal elasticity was obtained for 25% mass share of the modifier.

As predicted, the greater the mass fraction of the alundum grains, the greater the E value. The E modulus also increases with the decrease in grain size (higher F). The biggest difference against the reference samples was observed for the EA F360 composite, amounting to 2788.44 MPa (an increase in the material’s modulus of elasticity by 26% with regard to samples made of pure resin). Thereby, the smallest difference with regard to the reference samples was observed in the case of 5% mass share of EA F220 in the composite.

On the basis of Figure 7, it was observed that the highest values of max principal stresses of the tested samples in most cases were obtained for 10% share of alundum (F240, F280, F320, F360). The highest obtained value of 62.31 MPa was achieved for the EA F360 (130% more than for the reference samples). Similar conclusions were drawn on the basis of Figure 8, which indicates that EA F240, F280, F320, and F360, at a level of 10% mass share, ensured the largest elongation with regard to other percentage mass share values. In the case of 10% EA F360 share, the elongation against the reference sample differed by 103%. It should be mentioned that in most cases, the lowest elongation values were obtained for composites containing 25% mass shares of the friction modifier. The obtained results show that it is absolutely crucial to determine the effect of relationships between the composition of polymeric composites and the obtained properties due to a non-linear course of formation of the selected mechanical properties. Based on these test results, it can be concluded that 10% of EA is the critical concentration at which the maximum tensile strength occurs. A further increase in the mass fraction of EA, regardless of the grain size, causes a decrease in strength. It can be justified by the presence of sharp edges in the grains of EA and, at the same time, a smaller distance between the grains in the volume of the composite, and thus a smaller and smaller surface of the resin subject to damage due to the lateral approach of the grain edges during axial stretching of the composite samples.

## 4. Conclusions

On the basis of the conducted research, it was found that along with a rise of the average value of weight loss, the weight loss between the cycles was also growing. When comparing composite wear with an addition of various grains, it was found that alundum of grain sizes equal to F220 and F240 exerted the biggest influence on the reduction of abrasive wear of the tested samples. However, taking into account the roughness parameters, the F280 alundum had the effect of obtaining the lowest roughness parameters, Ra approximately at level 1 µm, and Rmax averaged 76 µm for a mass share of 15%. Regardless of the mass share for these grains, the abrasive wear was the lowest after each cycle interval (in the case of F220, on average it equalled 14.04% compared to the average value of the weight loss between the cycles for all percentage shares of the exploited grains). The addition of a physical modifier of friction influenced the selected mechanical properties of the composites, both in the case of the value of the modulus of longitudinal elasticity and the max principal stresses value. The highest value of the material’s modulus of longitudinal elasticity was obtained for 25% of the mass share of alundum. Taking into account the differences against the reference samples, it was observed that for the EA F360 composite, it is the largest, amounting to 2788.44 MPa, with an increase in the material’s modulus of longitudinal elasticity by 26% in relation to samples made of pure resin. The smallest effect of the presence of the additive against the reference samples was observed in the case of 10% mass share of EA F220 in the composite (an increase by merely 0.7%). However, no relationships were observed between them and the obtained hardness of the composites. The obtained results indicate a lack of linear relationships occurring between the observed properties and the composition of the composites. Currently, the authors are conducting research aimed at making multi-criteria optimisation of composites’ composition containing alundum with epoxy matrix base.

## Figures and Tables

**Figure 1 materials-13-05735-f001:**
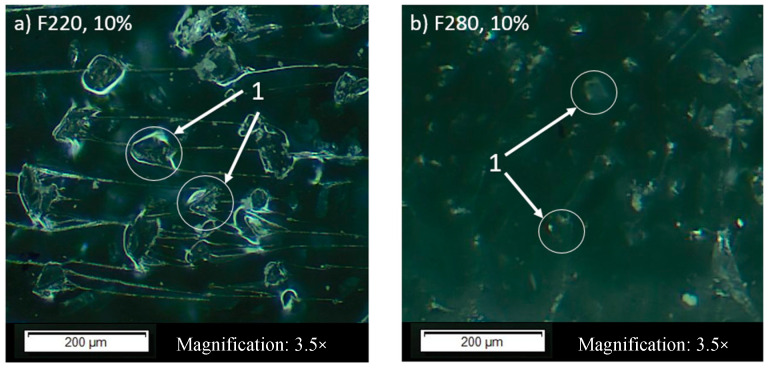
Fracture surfaces of epoxy composites with 10% mass share of alundum, (**a**) F220, (**b**) F280. 1—grain of alundum.

**Figure 2 materials-13-05735-f002:**
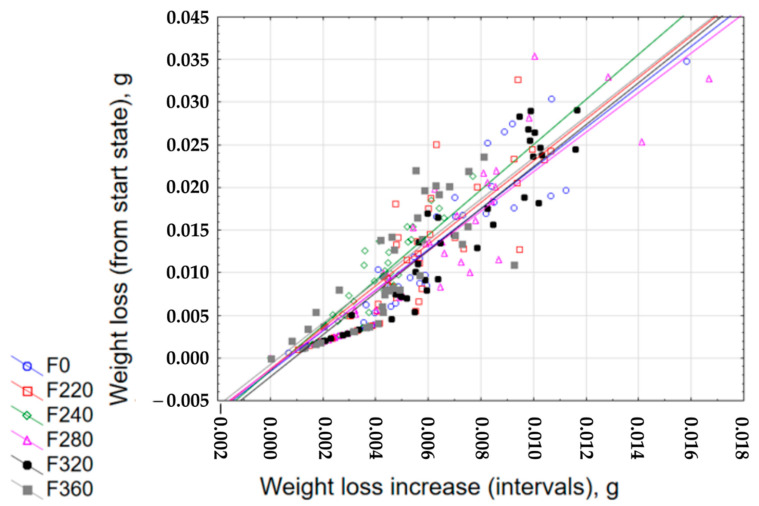
Exemplary graphs of the loss of weight discrepancy from the initial state, with regard to the weight loss between the cycles for epoxy resin with 5% addition of alundum with the following grains.

**Figure 3 materials-13-05735-f003:**
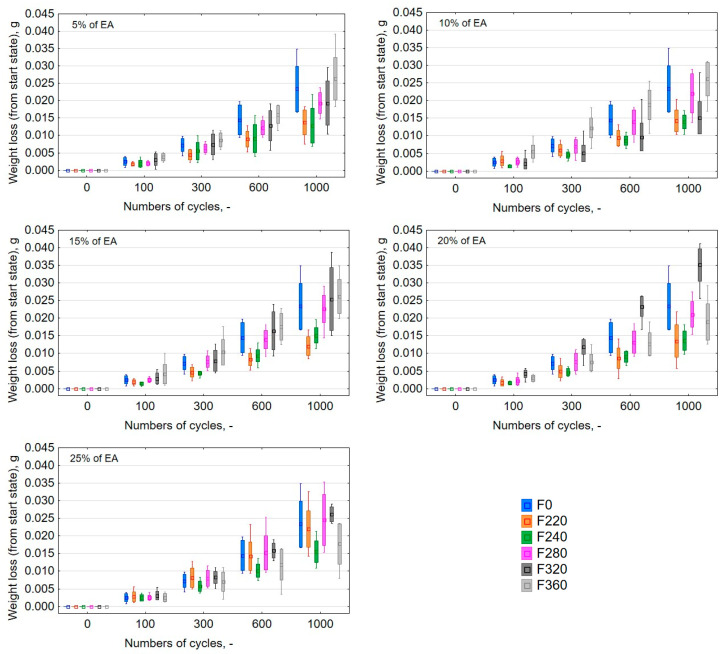
Weight loss graphs against the number of cycles for samples made with L285 resin and H285 hardener with an addition of alundum with the following grains: F220, F240, F280, F320, and F360 for percentage mass share of the physical friction modifier equal to 5%, 10%, 15%, 20%, and 25%, respectively.

**Figure 4 materials-13-05735-f004:**
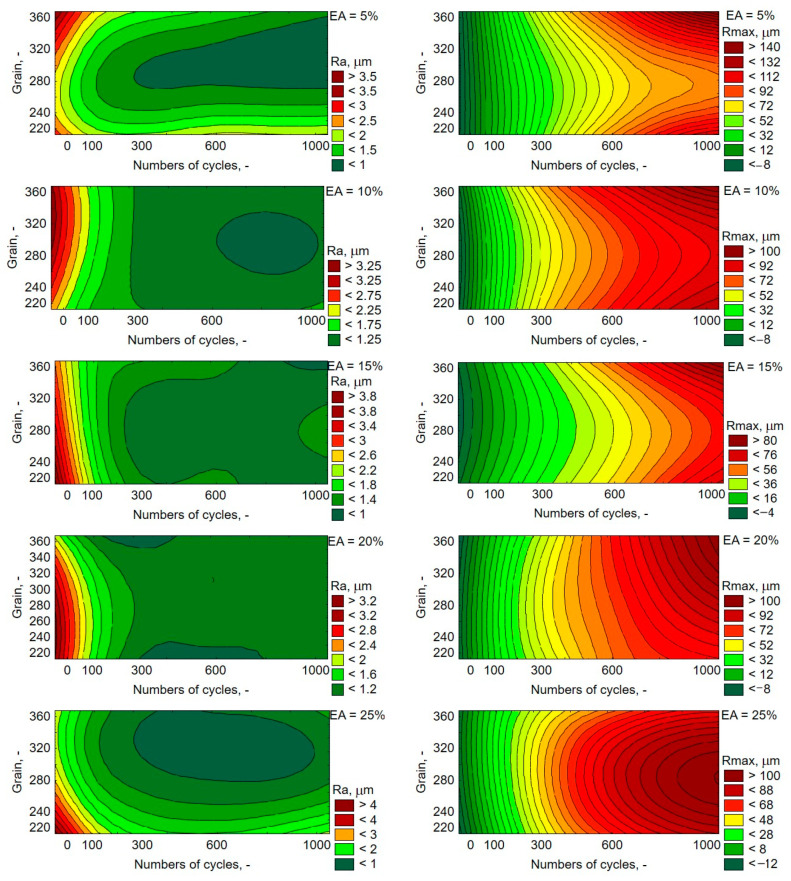
Characteristics of changes in parameters Ra and Rmax, depending on the number of friction cycles and grain of alundum.

**Figure 5 materials-13-05735-f005:**
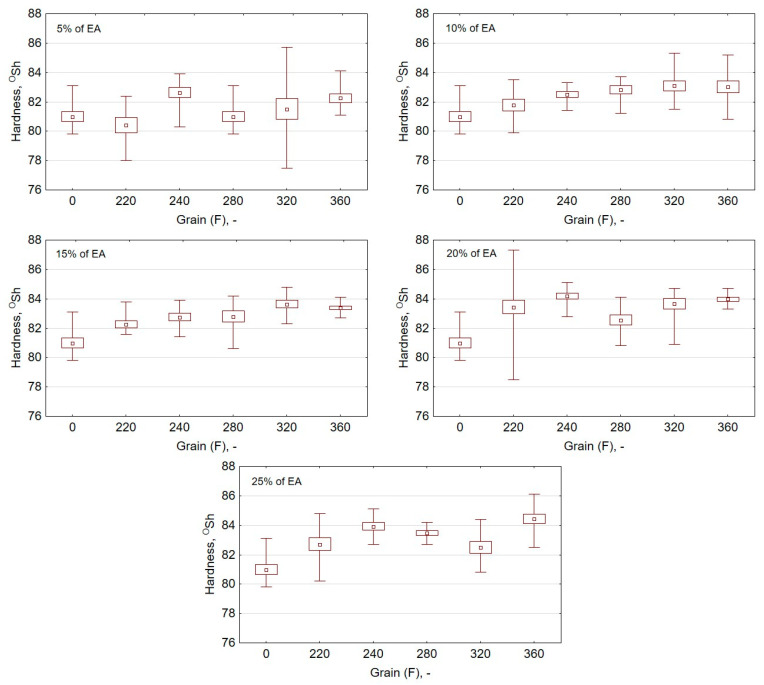
Graphs of values of hardness measurements by means of the Shore method for samples with an addition of alundum of different grain for mass percentage share of the physical friction modifier equal to 5, 10, 15, 20, and 25%, respectively.

**Figure 6 materials-13-05735-f006:**
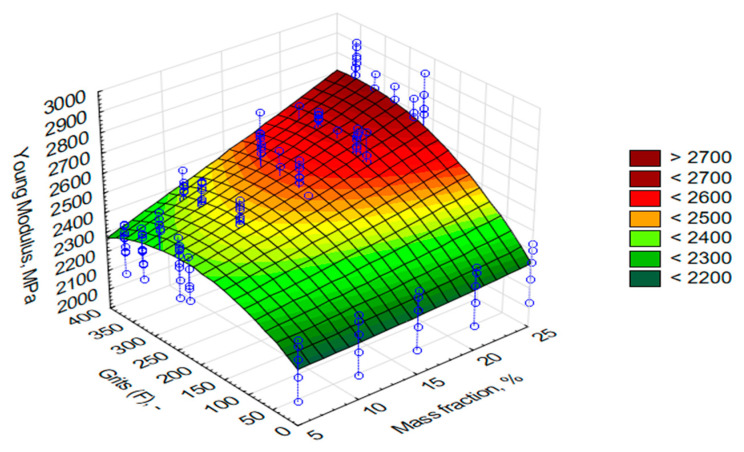
Graph of the modulus of elasticity values of the material for samples made with L285 resin and H285 hardener with an addition of alundum with the following grains: F220, F240, F280, F320, and F360, with various percentage mass shares.

**Figure 7 materials-13-05735-f007:**
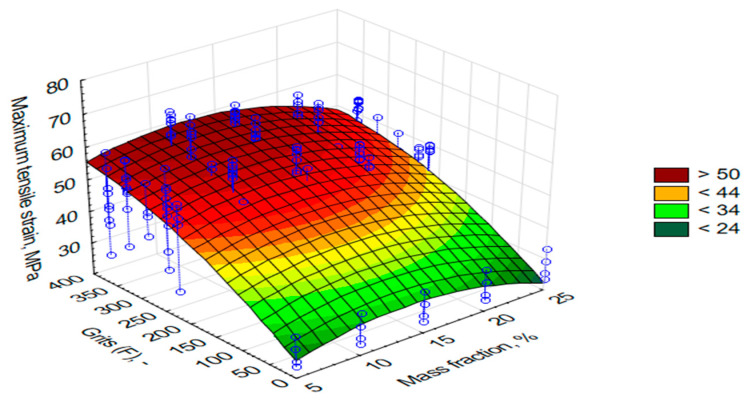
Graph of max principal stress values for samples made with L285 resin and H285 hardener with an addition of alundum with the following grains: F220, F240, F280, F320, and F360, with various percentage mass shares.

**Figure 8 materials-13-05735-f008:**
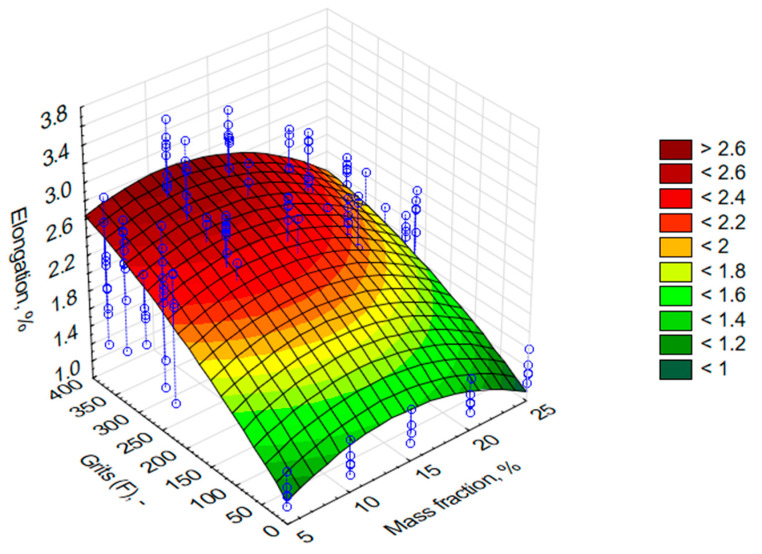
Graph of relative elongation values made with L285 resin and H285 hardener with an addition of alundum with the following grains: F220, F240, F280, F320, and F360, with different various percentage mass shares.

**Table 1 materials-13-05735-t001:** A list of exemplary scientific studies on the influence of modifiers on the properties of composites [37,38,39,40,41,42,43].

Modifier	Short Description	Results	Reference
graphite and talc particles	Katiyar with the team researched the impact of different weight percentages (wt%) of graphite and talc particles to SU-8 (an epoxy-based polymer). They stated that the composite with the optimized composition of SU-8 + 15 wt% graphite + 15 wt% talc has shown superior properties compared with pure SU-8 and other tested composites.	Four times lesser steady-state coefficient of friction (~0.2), three times higher elastic modulus (~7.97 GPa), two times greater hardness (~0.52 GPa) over those of pure SU-8.	[37]
graphite and/or carbon nanotubes	Researchers determined the impact of different fillers on the tribological behaviour of an epoxy (treated and untreated carbon nanotubes, graphite, and a mixture of graphite and carbon nanotubes).	The best result was obtained with TCNTs (tapered carbon nanotubes), which were well dispersed in the epoxy matrix due to the presence of the NH_2_ groups.	[38]
liquid lubricants and graphene or graphite particles	Epoxy and its composites filled with liquid lubricants and graphene or graphite particles were coated onto a D2 tool cylindrical steel shaft (the thickness of the coating was equal to 50–60 μm). Base-oil SN150 of Group-I and PFPE, graphene with particle sizes <100 nm and graphite with particle sizes <20 μm were used. The epoxy used was Araldite AY 103 with HY 951 hardener.	2 times reduction coefficient of friction over the sample without liquid filler; the epoxy/SN150 gives the lowest CoF (0.045), but at high load epoxy/graphene/SN150 shows the lowest coefficient of friction.	[39]
lamellar-structure expanded graphite (nano-EG)	The authors investigated the effects of adding nanoscale lamellar-structure expanded graphite (nano-EG) on the friction and wear properties of hot-moulded polyimide (PI)-based composites. The friction and wear tests were carried out using a Type 1045 steel ring, rotating against a composite disk.	The best tribological properties occurred when the nano-EG content is 15 wt% (wear resistance of PI/nano-EG nanocomposite increased 200 times).	[40]
glassy carbon	The research of the glassy carbon on the properties of the composites was performed for a material with a polymeric and metallic matrix. The carbon particles had a diameter of approx. 100 µm and their *w/w* concentration varied from 2 to 10% (polymeric matrix).	After adding glassy carbon (4–6%), the friction coefficient falls by approx. 10%. The presence of glassy carbon contributes to the stabilisation of the friction coefficient in the temperature range of 280–350 °C. With large amounts of glassy carbon (ca. 8–10%) the friction coefficient reduction in a critical temperature range is insignificant.	[41]
carbon nanotubes	The authors prepared nanocomposites containing 1 wt% of carbon nanotubes (CNTs) with different lengths. They studied using a block-on-ring tribometer at different sliding velocities.	The CNT addition increased storage modulus, loss modulus, and thermal conductivity of the composites. It is possible to influence their tribological properties.	[42]
multi-walled carbon nanotubes	Zhang et al. studied the effect of pristine multi-walled CNTs on wear resistance of their epoxy composites. They used multi-walled carbon nanotubes (MWCNs) prepared by chemical vapour deposition with their diameters ranging from 10 to 25 nm and their lengths from 10 to 20 μm. Dry wear tests of the CNT–epoxy composites were carried out by the authors on a pin-on-disc machine.	It was found that the surface coverage area of CNTs plays a significant role in the wearability of the polymer composites. When R_c/m_ is bigger than 25%, the wear rate can be reduced by a factor of 5.5.	[43]

**Table 2 materials-13-05735-t002:** Regression functions of the loss of weight discrepancy from the initial state, with regard to the loss of weight between the cycles for epoxy resin with various mass shares and grains.

Abrasive Grain	Percentage Mass Share	y = b + ax	r^2^
EA F220	5%	y = −0.0004 + 2.2461x	0.8042
	10%	y = −0.0004 + 2.4164x	0.7383
	15%	y = −0.0004 + 2.4136x	0.7311
	20%	y = −0.0007 + 2.3873x	0.8388
	25%	y = −0.0013 + 2.4469x	0.8067
EA F240	5%	y = −0.0003 + 2.402x	0.6845
	10%	y = −0.0012 + 2.4619x	0.8865
	15%	y = −0.0011 + 2.3703x	0.8884
	20%	y = −0.0008 + 2.4045x	0.8113
	25%	y = −0.0015 + 2.6468x	0.8531
EA F280	5%	y = −0.0018 + 2.524x	0.0886
	10%	y = −0.0019 + 2.4911x	0.8975
	15%	y = −0.002 + 2.5077x	0.9011
	20%	y = −0.001 + 2.3327x	0.8070
	25%	y = −0.0013 + 2.3209x	0.8644
EA F320	5%	y = −0.0015 + 2.6024x	0.8228
	10%	y = −0.0012 + 2.5039x	0.8378
	15%	y = −0.0019 + 2.4315x	0.8858
	20%	y = −0.0033 + 2.596x	0.8490
	25%	y = −0.0022 + 2.4659x	0.9001
EA F360	5%	y = −0.0001 + 2.111x	0.8212
	10%	y = −0.0002 + 2.43x	0.5933
	15%	y = −0.0012 + 2.4442x	0.7298
	20%	y = −0.001 + 2.4708x	0.7829
	25%	y = −0.0008 + 2.424x	0.7581

**Table 3 materials-13-05735-t003:** Density of samples made of L285 resin and H285 hardener with an addition of alundum with the following grains: F220, F240, F280, F320, and F360, with various percentage mass shares.

F220		F240		F280		F320		F360	
Share (%)	Density (g/cm^3^)	Share (%)	Density (g/cm^3^)	Share (%)	Density (g/cm^3^)	Share (%)	Density (g/cm^3^)	Share (%)	Density (g/cm^3^)
0	1.173	0	1.173	0	1.173	0	1.173	0	1.173
5	1.216	5	1.213	5	1.213	5	1.212	5	1.209
10	1.262	10	1.261	10	1.274	10	1.260	10	1.257
15	1.319	15	1.313	15	1.334	15	1.314	15	1.307
20	1.349	20	1.357	20	1.372	20	1.366	20	1.350
25	1.446	25	1.414	25	1.429	25	1.418	25	1.410

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
