# Peer review of "Research into the Effect of Grain and the Content of Alundum on Tribological Properties and Selected Mechanical Properties of Polymer Composites"

_materials, 2020, doi:10.3390/ma13245735_

Round 1
Reviewer 1 Report
The authors study the effects on the tribological and mechanical properties of polymer composites. The organization of the paper is ok and the text is clear.
However, more detailed information is needed.
For example, the paper need validation by either experiment or other published data. More detailed information is required in the introduction section.
For example, Line 104 - 105: please summarize or explain the literature a little bit instead of only listing the references. Also for Line 154: Figure 5.1 should be Figure 1
In addition, the authors could compare or validate their results with some published data in the results. For example, Line 263 to 273, please explain the findings, i.e., what is the meaning of the data?
Is there any reference supporting the data? Please also address the implications for this paper in the conclusion section.
Author Response
Dear Reviewer,
We would like to thank you for suggestions that helped to improve the description of the methods and the discussion of the results. Below we present our answers. In the text, all changes are highlight yellow.
Comment
For example, Line 104 - 105: please summarize or explain the literature a little bit instead of only listing the references. Also for Line 154: Figure 5.1 should be Figure 1.
Response
Thank you for your comments. We summarized the literature shown in the first part of the article. We changed the number of the Figure 5.1 too.
Comment
In addition, the authors could compare or validate their results with some published data in the results. For example, Line 263 to 273, please explain the findings, i.e., what is the meaning of the data?
Is there any reference supporting the data? Please also address the implications for this paper in the conclusion section.
Response
We have modified the content of the article in several places. Please see if the changes are appropriate in order to better understand the research results.
Thank you very much for your advice and help. We want this article to be the starting point for the next steps. We will soon write a new publication that will present the studied phenomena from a different perspective.
Kind regards
Authors
Reviewer 2 Report
The authors report about the tribological characterization of a polymer matrix composite made of epoxy resin and alundum for different grain sizes and mass percentage shares of the filler. They demonstrated the dependency of the abrasive wear, roughness, elastic modulus, tensile strain, hardness and density from both the alundum content and grain size.
In general, the article is well organized and written. However, it appears to be more like a technical report than a research article as the discussion of the results is limited only to describing the obtained data without giving any explanation of those findings. Moreover, some additional details are required about the experimental approach.
In the following the authors can find some comments/suggestions that could help to improve the description of the methods and the discussion of the results.
Major comments:
Line 121: it would be useful to add a table containing the comparison between the FEPA definition of the grain sizes and the actual grain sizes.
Line 125: the authors used the gravity casting method to produce the composites without providing any information about the operational parameters and the process itself. How do they control the dispersion of the filler within the matrix? Is it homogeneously dispersed? Please, add some details about the process.
Line 128: the authors declare to prepare 10 samples. Please, clarify that each experimental scenario has been replicated 10 times. However, did the authors produce 10 different composites for each combination of alundum content and grain size or just one for each of them and then they cut 10 samples from the laminate?
Lines 141-142: the authors state to observe the wear surface with an optical microscope, but no images are attached within the manuscript. Please, add some of them in order to better clarify the effect of the alundum content and grain size on the tribological properties.
Lines 174-181: why a greater grain size together with the lower values of mass share result in a greater weight loss and abrasive wear? Is it due to the removal of alundum particles which leaves big pores? Please, clarify and add some optical images (according to the previous comment). This should be helpful also in explaining the effect on the roughness.
Lines 246 and 264: How do the authors explain the difference in the elastic modulus and tensile strain? Why does the first is greater for F360 and 25% while the other for F360 but 10%?
Minor comments:
Lines 149-150: was the hardness tester used for both hardness and density measurements? Please, clarify.
Line 154: Reference to Figure 1 is wrong. Please, correct.
Table 1: please, rearrange the table by avoiding repeating the equation and the r2 term for each combination maybe indicating just "a" and "b". Moreover, the authors can also use only the designations F220, F240, etc. instead of L285+H285+...
Lines 211 and 213: is the unit of measure of the hardness correct?
Figure 6: along the z-axis is written tensile "strein". Please, modify in tensile "strain".
Author Response
Dear Reviewer,
We would like to thank you for suggestions that helped to improve the description of the methods and the discussion of the results. Below we present our answers. In the text, all changes are highlight yellow.
Comment
Line 121: it would be useful to add a table containing the comparison between the FEPA definition of the grain sizes and the actual grain sizes.
Response
Thank you for this suggestion. We have included a sentence containing the comparison between the FEPA definition of the grain sizes and the actual grain sizes.
Comment
Line 125: the authors used the gravity casting method to produce the composites without providing any information about the operational parameters and the process itself. How do they control the dispersion of the filler within the matrix? Is it homogeneously dispersed? Please, add some details about the process.
Response
We added some sentences and images in part 2 of the manuscript.
Comment
Line 128: the authors declare to prepare 10 samples. Please, clarify that each experimental scenario has been replicated 10 times. However, did the authors produce 10 different composites for each combination of alundum content and grain size or just one for each of them and then they cut 10 samples from the laminate?
Response
We produced 10 different composites for each combination of alundum content and grain size or just one for each of them (250 samples with alundum and 10 samples without alundum). Each of results on Figures are the average of 10 data. We were modified some sentences.
Comment
Lines 141-142: the authors state to observe the wear surface with an optical microscope, but no images are attached within the manuscript. Please, add some of them to better clarify the effect of the alundum content and grain size on the tribological properties.
Response
We added some sentences and images in part 2 of the manuscript.
Comment
Lines 174-181: why a greater grain size together with the lower values of mass share result in a greater weight loss and abrasive wear? Is it due to the removal of alundum particles which leaves big pores? Please, clarify and add some optical images (according to the previous comment). This should be helpful also in explaining the effect on the roughness.
Response
Alundum named F360 means grains with the smallest size tested. We added this explanation in the text on line 127 (numbering in the revised manuscript). We also added an explanation of the effect of grain size on weight loss and roughness in the text from verse 191 and verse 219.
Comment
Lines 246 and 264: How do the authors explain the difference in the elastic modulus and tensile strain? Why does the first is greater for F360 and 25% while the other for F360 but 10%?
Response
Was it about F220 at the end of the sentence? We modified the text, we included the smallest percentage content od EA.
Comment
Lines 149-150: was the hardness tester used for both hardness and density measurements? Please, clarify.
Response
We sorry, it was a mistake. We used the hardness tester Digi Test II, Type DTAA Bareiss for hardness measurements and Mettler Toledo XSE205DU/M Analytical Balance for density measurements.
Comment
Line 154: Reference to Figure 1 is wrong. Please, correct.
Response
Thank you for your remark. We changed reference and now is correct.
Comment
Table 1: please, rearrange the table by avoiding repeating the equation and the r2 term for each combination maybe indicating just "a" and "b". Moreover, the authors can also use only the designations F220, F240, etc. instead of L285+H285+...
Response
We had rearranged the table by avoiding repeating the equation and the r2 term for each combination. We hope that actually table looks clarity.
Comment
Lines 211 and 213: is the unit of measure of the hardness correct?
Response
Thank you for catching this error. We have changed the unit of the hardness (⁰Sh).
Comment
Figure 6: along the z-axis is written tensile "strein". Please, modify in tensile "strain".
Response
Thank you for noticing the typo. We have modified name of the z-axis and changed “strein” on “strain”.
Thank you very much for your advice and help. We want this article to be the starting point for the next steps. We will soon write a new publication that will present the studied phenomena from a different perspective.
Kind regards
Authors
Round 2
Reviewer 1 Report
The authors have address all of my comments.
Please address the error of the reference [38]: Line 106 [38, error!], and the errors in the last two rows of Table 1.
Author Response
Dear Reviewer,
We would like to thank you for your review. We checked your suggestions and made the necessary corrections. We made corrections in numbering of the references. We apologize for the oversight.
Kind regards
Authors
Reviewer 2 Report
The authors improved the overall quality of the manuscript by carefully following and accommodating all the reviewers' comments and suggestions. Well done.
Only one minor comment:
please, verify the numbering of the references at Line 107 and within the next Table 1.
Author Response

(The authors gave the same response as above.)
